# Long-Term Treatment of Lipoprotein Lipase Deficiency with Medium-Chain Triglyceride-Enriched Diet: A Case Series

**DOI:** 10.3390/nu15163535

**Published:** 2023-08-11

**Authors:** Liali Aljouda, Laura Nagy, Andreas Schulze

**Affiliations:** 1Clinical and Metabolic Genetics, The Hospital for Sick Children, Toronto, ON M5G 1E8, Canadalaura.nagy@sickkids.ca (L.N.); 2Department of Pediatrics, University of Toronto, Toronto, ON M5S 1A1, Canada; 3Department of Biochemistry, University of Toronto, Toronto, ON M5S 1A1, Canada

**Keywords:** lipoprotein lipase deficiency, MCT diet, inborn errors of metabolism

## Abstract

**Background**: Lipoprotein lipase (LPL) deficiency is a genetic condition. Affected individuals typically develop symptoms related to severe and persistent hypertriglyceridemia, such as abdominal pain and recurrent pancreatitis, before 10 years of age. No pharmacological treatment sustainably lowering triglycerides (TGs) in LPL deficiency patients has been proven to be effective. This study investigated whether a long-chain triglyceride (LCT)-restricted, medium-chain triglyceride (MCT)-supplemented diet enables a meaningful reduction in TGs and reduces LPL-related symptoms in children with LPL deficiency. **Methods:** A single-center retrospective case series study of LPL deficiency patients treated at the Hospital of Sick Children between January 2000 and December 2022 was carried out. Data, extracted from hospital charts, included demographics, diagnosis confirmation, clinical and imaging observations, and biochemical profiles. **Results:** Seven patients with hypertriglyceridemia > 20 mmol/L suspected of an LPL deficiency diagnosis were included. Six patients had a confirmed molecular diagnosis of LPL deficiency, and one had glycogen storage disease type 1a (GSD1a). Clinical presentation was at a median of 30 days of age (range 1–105), and treatment start, excluding one late-treated patient, was at a median of 42 days (range 2–106). The observation and treatment period of the LPL patients was 48.0 patient years (median 7.1, range 4.3–15.5). The LCT-restricted and MCT-supplemented diet led to an immediate drop in TGs in six out of six LPL patients. TGs improved from a median of 40.9 mmol/L (range 11.4–276.5) pre-treatment to a median of 12.0 mmol/L (range 1.1–36.6) during treatment, total cholesterol from 7.6 mmol/L (4.9–27.0) to 3.9 mmol/L (1.7–8.2), and pancreatic lipase from 631 IU/L (30–1200) to 26.5 IU/L (5–289). In 48 patient years, there was only one complication of pancreatitis and no other disease-specific manifestations or complications. Catch-up growth was observed in one late-treated patient. All patients maintained normal growth and development. As expected, the diet failed to treat hypertriglyceridemia in the GSD1a patient. **Conclusions:** The dietary restriction of LCT in combination with MCT supplementation as long-term management of pediatric patients with LPL deficiency was feasible, well tolerated, and clinically effective in reducing TG levels and in preventing LPL-related complications.

## 1. Introduction

Lipoprotein lipase (LPL) deficiency is a rare autosomal recessive condition due to biallelic pathogenic variants in the LPL gene that disrupts the postabsorptive conversion of alimentary fats [1]. Lipoprotein metabolism, especially the metabolization of chylomicrons and very-low-density lipoproteins, relies on the function of the enzyme LPL. A consequence of enzyme dysfunction is a distinct rise in serum triglycerides (TGs). 

LPL deficiency usually presents in childhood with episodes of abdominal pain and recurrent episodes of acute pancreatitis requiring hospitalization. Left untreated, pancreatitis can develop into a chronic condition with the permanent destruction of pancreatic cells, which results in exocrine and endocrine pancreatic insufficiency. Additionally, patients with LPL deficiency may manifest eruptive cutaneous xanthomata, lipemia retinalis, and hepatosplenomegaly. Affected individuals typically develop signs and symptoms before 10 years of age, with one-quarter having the first symptoms at one year of age [2]. 

The primary long-term treatment for LPL (lipoprotein lipase) has traditionally been dietary modification, specifically a low-fat diet with a focus on reducing long-chain TGs (LCTs). Other past and present treatment options include *LPL* gene therapies (alipogene tiparvovec), which involve introducing functional copies of the LPL gene using adeno-associated viral vector 1 [1], diacylglycerol acyltransferase 1 inhibition [3], inhibitors of the microsomal TG transfer protein [4], angiopoietin-like protein 3 [5], and apoC-III [6]. To date, these treatment options have either not been proven successful in lowering TG levels in LPL patients, have had to be discontinued for various reasons, or are still under investigation. Currently, there are no Food and Drug Administration-approved medications, and lipid-lowering drugs have limited or no success [7,8,9].

Treatment with medium-chain TGs (MCTs) has been proposed since MCTs are not incorporated into chylomicron bodies and are absorbed directly from the portal vein [10]. However, to the best of our knowledge, the treatment of LPL patients with an MCT-enriched diet has only been reported in one single-center retrospective study of fifteen patients with hypertriglyceridemia above 11.3 mmol/L. The diet was fat-restricted and supplemented with MCT oil to comprise 10% of the total caloric requirement, and it led to a drastic reduction in the fasting lipid profile, TGs, cholesterol, and VLDL after 24 days of treatment [11]. It is known that the provision of MCT is of particular importance for infants and children to ensure adequate fat intake for growth and development. Furthermore, the treatment of long-chain fatty acid oxidation defects with an LCT-restricted/MCT-enriched diet has become a successful intervention that changes the natural course of the diseases [12] and that is being practiced in all metabolic centers for inborn errors of metabolism in the world.

For more than a decade, the metabolic service at the Hospital for Sick Children has had success in the treatment of LPL patients with an LCT-restricted/MCT-enriched diet. Currently, such dietary modifications are not the standard of care for LPL deficiency or hypertriglyceridemia. This study investigated whether a long-chain triglyceride (LCT)-restricted, medium-chain triglyceride (MCT)-supplemented diet enables a meaningful reduction in TGs and reduces LPL-related symptoms in children with LPL deficiency. 

## 2. Methods

In a single-center, retrospective, case review study, patients with hypertriglyceridemia above 20 mmol/L who have been or are still being treated with an MCT-enriched diet by the metabolic service of the Hospital for Sick Children were included. After receiving parental consent, data were abstracted from medical records and recorded in the study database. The chart review included charts from 1 January 2000 to 1 December 2022 of male and female patients below 18 years of age.

We were able to enroll seven patients, six female and one male, with hypertriglyceridemia. We report the clinical manifestations and molecular findings in these patients. The response to an LCT-restricted/MCT-enriched diet was monitored with respect to the clinical course, disease-specific complications, and biochemical profile—specifically pancreatic lipase (reference range 4–39 IU/L), amylase (reference range < 102 IU/L), TGs (reference ranges < 0.85 mmol/L (0–12 years) and < 1.02 mmol/L (12 years and older)), cholesterol (reference range < 4.4 mmol/L), and essential fatty acids (EFAs).

This study was approved by the Research Ethics Board of The Hospital for Sick Children, Toronto, Canada. 

### Red Blood Cell (RBC) Total Lipid Fatty Acid Profile

RBCs were separated from 1.5–3 mL EDTA blood, washed with phosphate-buffered saline, and transferred to screw-capped plastic (low in phthalate) tubes such as NUNC micro screw-top freezer tubes and were flushed with a stream of nitrogen for about 10 s before being capped tightly and stored at −80 °C. The samples were analyzed at the Peroxisomal Diseases Laboratory at the Kennedy Krieger Institute in Baltimore, Maryland. RBC membrane levels of fatty acids (C8–C26) were determined using capillary gas chromatography–mass spectroscopy of pentafluorobenzyl bromide fatty acid methyl esters [13].

## 3. Patients

### 3.1. Patient 1

This girl is the first child of consanguineous parents of Indian descent, with an uneventful antenatal and perinatal history. At 40 days of age, she presented with a one-day history of persistent emesis, not tolerating her feeds. An abdominal ultrasound demonstrated a mass in the head of the pancreas, for which she was transferred to a tertiary pediatric center for further management. 

The laboratory results at admission revealed a significant elevation in TGs to 276 mmol/L, cholesterol to 27 mmol/L, and pancreatic lipase to >1200 U/L. A complete blood cell count, electrolytes, transaminases, and liver synthetic function tests were within normal limits. She was admitted to the general pediatric ward for pancreatitis secondary to hypertriglyceridemia, raising the suspicion of a genetic dyslipidemia disorder. Upon admission, oral intake was stopped, and she was managed with I.V. fluids that led to the normalization of TGs. Re-establishing enteral nutrition caused a significant rise in TGs to 117 mmol/L and pancreatic lipase of 694 U/L. On examination, she had increased muscle tone. Due to the concern of blood hyper-viscosity, a brain MRI showed no evidence of infarction or clots; she was transferred to the PICU, where she received two days of plasmapheresis until the normalization of TGs.

The initial suspicion of pancreatic mass was ruled out, as an abdominal MRI showed pancreatic inflammation consistent with pancreatitis. 

After the discontinuation of plasmapheresis, the metabolic service was consulted, and she started on an MCT-enriched formula (Lipistart), with her TGs showing a downward trend and stabilizing below 10 mmol/L until discharge. 

Targeted genetic testing confirmed the LPL deficiency diagnosis due to a homozygous pathogenic variant in the *LPL* gene (c. 1014 C>A, p. Tyr338*). Both parents were heterozygous for the familial variant. 

After discharge, she was maintained on an MCT diet with good tolerance. She was fed Lipistart^®^ (Vitaflo, Alexandria, VA, USA) exclusively for the first six months of her life. Regular blood work over four years showed stable TGs within our target range for LPL patients of <10 mmol/L. In addition, she had excellent weight gain, linear growth, up-to-age developmental milestones, and no further episodes of pancreatitis or other LPL-deficiency-related complications. 

### 3.2. Patient 2-I 

This girl is the first child of consanguineous, healthy parents of Pakistani descent. Pre-, peri-, and post-natal periods were unremarkable. At 3 ½ months of age, isolated splenomegaly was noted at a routine well-baby checkup. Blood work unmasked significant hypertriglyceridemia of > 50 mml/L; thus, she was referred to our center. Repeat labs confirmed hypertriglyceridemia of 29 mmol/L and normal cholesterol, amylase, and pancreatic lipase. She was started on an LCT-restricted/MCT-enriched formula with Lipistart^®^ and Portagen^®^ (Mead Johnson, Evansville, IN, USA). Molecular testing confirmed LPL deficiency with a homozygous pathogenic variant in the *LPL* gene (c.784C>T, p.Q262X) that segregated with the parents. 

By 18 months of age, she was growing and developing well. Her TG levels up to this age were between 15 and 18 mmol/L, which we considered protective against pancreatitis, but they rose to 20 mmol/L at 30 months of life. Changes were made, including discontinuing the Portagen^®^ formula, increasing MCT provision by adding MCT Procal^®^ (Vitaflo, Alexandria, VA, USA) to Lipistart^®^, and adding MCT oil to food. Moreover, EFA provision intake was suboptimal, for which the walnut oil dose as a source of EFAs was increased from 1.25 mL to 2.5 mL per day. Despite these changes, her TGs remained in the twenties, and her weight gain dropped. Incidentally, the unsatisfactory metabolic control and failure to thrive were related to unhealthy feeding habits, including a high number of snacks, no structured mealtimes, and feeding overnight. She was followed up by a pediatric psychiatrist expert for feeding challenges, after which her feeding habits improved and weight gain stabilized. Her TGs continued to fluctuate but lowered to 12–20 mmol/L. Until now, there have been no LPL-deficiency-related complications. 

### 3.3. Patient 2-II 

This child, the third of consanguineous healthy parents of Pakistani descent and a sibling of patient 2-I, was diagnosed due to a positive family history. After delivery, she was admitted to an outside NICU for feeding intolerance secondary to a cleft palate. Given the positive family history, she was tested for TGs on days two and six of life. Her TGs were 30 mmol/L, consistent with her diagnosis of LPL deficiency. She was started on an LCT-restricted/MCT-enriched formula (Lipistart^®^) on day 7 of life. By three months of age, she was noted to have severe global developmental delays, horizontal nystagmus, and facial dysmorphism. At the age of 4 years, she was diagnosed with an autistic spectrum disorder. Given consanguinity, a second genetic condition was suspected; therefore, she was referred to a genetics service for assessment. An extensive genetic workup, including trio whole-exome sequencing, was performed, which confirmed the diagnosis of LPL deficiency with the same homozygous variant as her sister and a homozygous variant of uncertain significance in the *NAA25* gene, a gene involved in protein N-terminal acetylation, which is a candidate gene with a potential relationship with the patient’s disease phenotype; however, this requires further evaluation. 

From a metabolic perspective, since birth, she has been doing very well, tolerating her LCT-restricted/MCT-enriched diet. She has not had any LPL-related complications. Her TG levels have been mostly in the treatment target range of <10 mmol/L. She continues to gain weight and grow along the usual percentiles.

### 3.4. Patient 3

This girl is the second child of consanguineous parents of Pakistani descent. Pregnancy was notable for fetal growth restriction. She was delivered at 41 weeks with a birth weight of 2.4 kg. She was noted to have poor weight gain during the neonatal period despite adequate nutrition. In addition, she had a positive newborn screening for congenital hypothyroidism, for which she was followed by endocrinology. 

Given a positive family history of a first-degree cousin with LPL deficiency (patient 2-I), she was assessed by the metabolic team at our center at seven weeks of age. Blood workup revealed elevated TGs of 116 mmol/L and cholesterol of 19 mmol/L, leading to admission for further evaluation and the initiation of dietary treatment. Amylase was normal, with no signs of pancreatitis. She was successfully switched from breast milk to an LCT-restricted/MCT-enriched formula (Lipistart) with a slow but persistent drop in TGs. The LPL diagnosis was confirmed with a homozygous pathogenic variant in the *LPL* gene *(*(c.784 C>T, p.Q262X), the same as in her cousins. Both parents were heterozygous for the familial variant in LPL. After discharge, she continued to have excellent tolerance to the MCT-based formula, and her TGs were well within our treatment target range of <10 mmol/L. 

By six months of age, she was noted to have failure to thrive, despite receiving appropriate calories for her age and adequate metabolic control, which was out of proportion to what was known for LPL deficiency. Furthermore, she was noted to have a global developmental delay. 

At the age of 7 months, given the clinical symptoms and the persistent elevation of TSH to around 10 mIU/L (reference range 0.5–4.77) despite normal FT4 of 13.1 pmol/L (reference range 10–23) and adequate metabolic control, she was diagnosed with thyroid hormone resistance and was started on L-thyroxine, initially with 25 µIU/day, which was later increased to 50 µIU/day. This led to significantly improved growth and a catch-up in development.

The girl had one episode of pancreatitis. At four years of age, after eating lots of ice cream a few days prior to her presentation, her pancreatic lipase was 289 U/L and TGs were 14 mmol/L at the time of presenting in clinic, but they were likely much higher in the days before. She has since continued to do very well with her MCT diet, and her TGs are maintained within the treatment target range of <10 mmol/L. There have been no other LPL-deficiency-related complications and no further concerns regarding weight gain and growth. She also continues to make appropriate gains from a developmental perspective. 

Considering the complexity of the clinical presentation, the consanguinity, and the appreciation of some distinct facial features, the genetics service initiated further workup. Microarray and brain MRI were normal. Whole-exome sequencing revealed a homozygous variant of unknown significance in *IGF1R*, in addition to the already known LPL variant.

### 3.5. Patient 4 

This patient is the child of a consanguineous union between first cousins of Pakistani descent. She was doing well until the age of 20 days, when she presented to the emergency department with a history of lethargy, poor feeding, irritability, and abdominal distension. The blood taken at this visit was noted to be creamy. An abdominal ultrasound showed splenomegaly. The initial lipid profile showed elevated cholesterol at 7.65 and TGs of 45.5 mmol/L. She was started on an LCT-restricted/MCT-enriched diet consisting of Portagen^®^, Enfamil^®^ (Mead Johnson, Evansville, IN, USA) plus added walnut oil, creating a 60% MCT mixture. Her TGs improved quickly. By day 4 of admission, her TGs were 1.34 mmol/L. The diagnosis of LPL deficiency was confirmed via molecular testing, which showed a homozygous variant in the *LPL* gene (c.472G>A, p.A158T). At 18 months, the family moved to the Middle East but returned to our center for yearly follow-ups until the last follow-up at 17 years of age. Since diagnosis, she has followed the MCT-enriched diet. Her TG levels fluctuated over the years, with most readings being between 13 and 17 mmol/L. Her parents attributed the fluctuation to difficulty in adhering to the diet, given that she spends much time with the extended family. At the ages of 10 years and 17 years, she had high TG levels of 33 and 25 mmol/L, respectively, attributed to dining out and eating high-fat food. Fortunately, she remained stable with no signs of pancreatitis. Her TGs improved immediately after resuming the MCT diet. Since birth, she has achieved appropriate developmental milestones and has grown appropriately. She has not had any LPL-deficiency-related complications. 

### 3.6. Patient 5

This boy is the child of a consanguineous union between first cousins of Pakistani descent. At two weeks of age, he presented for circumcision, when blood work was carried out, and high TGs were found. No further details are known. However, at that time, the family in Pakistan was told to avoid high-fat food. The family history is remarkable in that both parents had hyperlipidemia, and the father had bypass surgery by the age of 60 years. One paternal uncle has hyperlipidemia and is on medications but has no cardiovascular complaints. The family moved to Canada in 2006 when he was six years old. He was followed up in the cardiology lipid clinic at the Hospital for Sick Children. His initial lipid profile showed elevated TGs; he was started on a fat-restricted diet and fibrate medication (Lipidil^®^). The diagnosis of LPL deficiency was confirmed via molecular testing, which showed a homozygous nonsense mutation in the *LPL* gene (Q262X, c.784C>T). The family’s compliance with the diet was insufficient. He continued to have an extremely elevated TG level of 45 mml/L, which ultimately affected his growth and development significantly. By seven years of age, he developed eruptive xanthomata spreading over his body, ears, forearms, and legs. He had no previous history of hospitalization with acute pancreatitis; however, he had frequent episodes of headache and abdominal pain related to eating fatty food. 

At the age of 14 years, he was referred to our metabolic clinic, and his diet was adjusted to an LCT-restricted/MCT-enriched diet. Since implementing the dietary changes, his TGs and clinical status have improved. Specifically, his growth, weight gain, and head circumference improved, crossing percentiles. However, compliance with the MCT diet remained an issue until adulthood, which is reflected in his TG readings and growth parameters.

### 3.7. Patient 6

This girl is the child of a non-consanguineous union of healthy parents of European descent. She was reportedly in good health until five months of age, when she began to experience growth failure. At nine months, her length was just above the 3rd percentile, while her weight was tracking along the 50th percentile. At that time, her weight started decreasing, she began to experience increased work of breathing, and she had tachypnea. Workup revealed significantly elevated TGs of 113 mmol/L, cholesterol of 14.43 mmol/L, elevated lactate, elevated transaminases, and high anion gap metabolic acidosis. An abdominal ultrasound showed significant hepatomegaly. 

As the differential diagnosis for her highly elevated TGs included LPL deficiency, she was managed with an LCT-restricted/MCT-enriched diet; however, due to a slow improvement in the TGs, she required intravenous fluids and parenteral nutrition for five days before significant improvement in TGs and lactate. She remained clinically well with no evidence of pancreatitis. Once the TGs were below 40 mmol/L, oral feeding was re-started with an MCT formula (Lipistart^®^), and the TGs began to show a downward trend. A subsequent observation of recurrent hypoglycemia and pathognomonic lactate trends in response to fasting/feeding led to the suspicion of glycogen storage disease type 1a (GSD1a). Consequently, our treatment plan was changed to manage both LPL and GSD1a until genetic diagnosis. She was discharged home on frequent feeds, Lipistart^®^ formula, and cornstarch for glucose homeostasis. Her glycemic control with this regimen was reasonable. She gained weight and made good developmental progress. Her metabolic control improved, with TGs below 10 mmol/L, normal lactate, and normal uric acid. After a molecular confirmation of GSD1a deficiency via the segregation of compound heterozygous pathogenic variants in *G6PC* (c.980_982del p. Phe327del/c.563-3C>G p.?), the Lipistart^®^ formula was stopped, while GSD1a management continued.

## 4. Diet Principles

The diet restricts LCT to a degree that is comparable to the restriction for a severe long-chain fatty acid oxidation defect [12]. We aim to provide approximately 10–15% of energy in the diet from LCT and 20–30% of energy from MCT, depending on the age of the child and the fat intake required to sustain growth and weight gain. Total fatty acid needs are higher for infants, and, therefore, total energy from fat, or the % energy from MCT, is higher in the neonatal period. The diet aims to provide only enough LCT to prevent a deficiency of essential fatty acids and to ensure that the diet is tolerable.

Sources of MCT in the infant diet are MCT-enriched formulas, such as Monogen^®^ (Nutricia, Rockville, MD, USA) or Lipistart^®^. Our patients are typically maintained on an MCT-enriched formula into childhood, as it is also an important source of vitamins and minerals, some of which are at risk of deficiency in a diet low in LCT. In childhood, or when the MCT-enriched formula is no longer tolerated, MCT is provided in the diet in the form of medical-grade MCT oil (Nestle), which is added during cooking or to foods, or an MCT supplement, such as Liquigen^®^ (Nutricia) or MCT Procal^®^ (Vitaflo), which is added to a fat-free beverage or food.

LCTs are provided from a source of essential fatty acids in the infantile period and then, once the infant begins to eat food, in minimal amounts from food. Food choices are either fat-free or very low in fat. Parents are often advised to count grams of fat in food and not to exceed the limit recommended by the dietitian.

The essential fatty acids linoleic acid and alpha-linolenic acid cannot be made by the body and therefore must be included in the diet in the minimum amounts reported to be required [12,14]. The inclusion of breastfeeding in the diet for infants is difficult due to the very high fat content of breastmilk, which is mostly LCT.

## 5. Results

This study included seven patients, six with LPL and one with GSD1a. The treatment- and observation period was 48.0 patient years, with a median of 7.1 and a range of 4.3–15.5 years (excluding patient # 7 who had a diagnosis of GSD1a and whose MCT-enriched diet stopped after the diagnosis). 

The six LPL patients were from four kindreds; all parents were consanguineous. Furthermore, three out of six had a family history of a similar condition, and five out of six were females (Table 1).

### 5.1. Clinical Presentation

The initial presentation was early, during the neonatal/early infantile period (median 30 days and range 1–105 days). The initial presentation of two cases (P1 and P4) was poor feeding, lethargy, and abdominal distension. In P1, abdominal distension was initially thought to be due to a pancreatic mass, ruled out after an abdominal MRI, which showed changes suggestive of pancreatitis. An abdominal US in P4 showed splenomegaly. Two patients (P 2-I and P5) were picked up during a routine checkup. Of these, P 2-I was found to have isolated splenomegaly, and the other was found to have elevated TGs in routine blood work. Given the family history of LPL deficiency in two cases (P2-II and P3), they were screened for LPL deficiency during the neonatal period (Table 1).

### 5.2. Treatment

Treatment with an LCT-restricted/MCT-enriched diet was implemented at a median age of 42 days (range 2–106), with the exception of patient P5, who was started at age 13.4 years. Our diet was initiated immediately after the initial presentation of four (P2-I, P2-II, P3, and P4) of the six cases. P1 was initially managed by the pediatric team. She underwent plasmapheresis, and our diet was started four days after her presentation. Patient P5 was initially followed by the cardiology lipid clinic. They tried diet and other interventions but could not lower TGs, after which he was referred to our service at the age of 13.4 years and started on an MCT-enriched diet. The diets of our patients are summarized in Table 2.

### 5.3. Biochemical Characteristics

An LCT-restricted and MCT-supplemented diet led to an immediate drop in TGs in six out of six LPL patients. TGs improved from a median of 40.9 mmol/L (range 11.4–276.5) pre-treatment to a median of 12.0 mmol/L (range 1.1–36.6) during treatment, total cholesterol decreased from 7.6 mmol/L (4.9–27.0) to 3.9 mmol/L (1.7–8.2), and pancreatic lipase decreased from 631 IU/L (30–1200) to 26.5 IU/L (5–289). 

All six patients had an immediate response and improvement in their TG level. Four (P1, P2-II, P3, and P4) of the six patients demonstrated plasma TG levels of < 10 mmol/L within five days of treatment or less (Figure 1). Due to unclear causes in P2-I, we could not establish TGs below 10 mmol/L; however, most of her readings were below 20 mmol/L, with no evidence of pancreatitis. In P5, routine blood work after initiating the MCT-enriched diet showed a significant decrease in TGs compared to the previous levels. Nonetheless, his readings were mainly around 20 mmol/L due to diet adherence issues.

Furthermore, amylase and pancreatic lipase were recorded in the patients’ charts during follow-up visits for all six patients, except for P4, for whom we did not have historical data on pancreatic lipase (Figure 1). We could not establish a clear correlation between pancreatic lipase, amylase, and TG levels. Pancreatic lipase was elevated in two patients who were symptomatic and presented with acute pancreatitis (P1 and P3). One child (P2-I) always had two times higher baseline pancreatic lipase than the other cases. However, she did not have any documented acute pancreatitis episode (Figure 1). Comparing the levels of TGs, cholesterol, and pancreatic lipase before initiating the MCT-enriched diet with the levels during the treatment observation period revealed a trend of improvement in all three parameters (Figure 2).

### 5.4. Complications

In 48 patient years, there was only one complication of pancreatitis and no other disease-specific manifestations or complications.

Patient P3 had a documented episode of acute pancreatitis, presenting with abdominal discomfort after eating a big portion of ice cream (pancreatic lipase 289 IU/L). Her TG concentration was in the treatment range when she reassumed the MCT-enriched diet the next day. She recovered without sequelae. Her metabolic control was excellent before and after the acute pancreatitis, with most readings being within the treatment target range of < 10 mmol/L.

Due to the uncontrolled TG levels preceding the initiation of the MCT-enriched diet, patient P5 had developed eruptive xanthoma in many parts of his body (ears, forearms, and legs). After the initiation of the MCT-enriched diet, his xanthomas abated over time, and there were no other documented LPL-related complications (Table 1).

We did not observe lipidemia retinalis, xanthomata, and organomegaly in any patient after the initiation of the LCT-restricted/CT-enriched diet.

### 5.5. Growth and Essential Fatty Acids

To assess the safety of the dietary intervention, the growth, development, and EFA levels of all six cases were followed very closely over the observation period (Figure 3). No persistent adverse effects on weight gain, linear growth, or pubertal development were seen in our patients. Patient P5 had catch-up growth after starting the MCT-enriched diet. In patient P3, initial growth failure was caused by hypothyroidism and was not diet-related. Once thyroxine supplementation was started, growth and development improved.

### 5.6. Genetic Variants

A genetic analysis was performed in all seven cases. All LPL patients had homozygous pathogenic variants in the *LPL* gene. Four were homozygous for the same pathogenic variant (p.Q262X). One patient was homozygous for p.Tyr338*, and one was homozygous for p.A158T (Table 1). We could not establish any genotype–phenotype correlation. Five of the pathogenic variants were missense mutations, and one was a nonsense mutation (Table 1). Two pathogenic variants in the *G6PC* gene (c.980_982del p. Phe327del/c.563-3C>G p.?) segregating with the parents were identified in patient 6 with GSD1a.

### 5.7. Imaging

An abdominal ultrasound was performed in four cases (P1, P3, P2-I, and P4). In two of them (P1 and P4), the abdominal ultrasound was conducted as part of the workup for the initial presentation. In one patient (P2-I), a follow-up abdominal ultrasound was carried out, as a routine checkup suspected splenomegaly in the physical examination. Lastly, in P3, the ultrasound was performed after she presented to the hospital with abdominal distention and elevated pancreatic lipase. Routine abdominal ultrasounds were not performed during the observation period. An abdominal and brain MRI was completed for P1 at initial presentation due to the suspicion of an abdominal mass in the ultrasound and an abnormal neurological examination. In patients P2-II and P3, a brain MRI was performed as part of the genetic workup of the medical complexity (Table 1).

## 6. Discussion

Lipoprotein lipase deficiency (LPL) is a rare autosomal recessive disorder characterized by impaired TG metabolism due to a mutation in the *LPL* gene. LPL deficiency is one of the major causes of familial chylomicronemia syndrome. In this retrospective chart review, we present the clinical characteristics and management outcomes of seven children presenting with hypertriglyceridemia >20 mmol/L, who were treated with an LCT-restricted/MCT-supplemented diet.

The clinical presentations of the patients in this study were consistent with the typical features of LPL deficiency. Some experienced episodes of abdominal pain and recurrent acute pancreatitis, a common LPL deficiency complication. The elevated serum TG levels observed in these patients are a hallmark of the disorder and can lead to various complications if left untreated, such as lipemia retinalis, eruptive cutaneous xanthomata, and hepatosplenomegaly. The association of LPL deficiency with an increased risk of cardiovascular disease is still controversial. The current literature does not consistently support an increased risk of cardiovascular disease in patients with LPL deficiency [15,16,17,18]. In one study, in four patients, peripheral or coronary atherosclerosis or both were observed before age 55 despite good metabolic control [16]. Hokanson et al. reported an increased risk of atherosclerosis with specific genotypes (Asp9Asn and Gly188Glu) and a cardioprotective effect with others (S447X) [17]. None of our patients had the high-risk genotypes described by Hokanson et al. [17]. Additionally, no cardiovascular disease or cardiac symptoms were observed during the observation period. Therefore, we did not routinely perform cardiovascular monitoring in our patients. Nevertheless, we cannot exclude the long-term risk of cardiovascular disease in individuals with LPL deficiency who maintain TG levels between 10 and 20 mmol/L. Long-term studies in adults would be necessary to assess the cardiovascular disease risk associated with TGs in this range.

LPL deficiency is the primary differential diagnosis to consider in the case of severe hypertriglyceridemia, i.e., TGs above 20 mmol/L. The clinical presentation of LPL can overlap with other conditions characterized by hypertriglyceridemia, such as uncontrolled diabetes mellitus, hypothyroidism, and specific medications like thiazide diuretics and estrogen-containing contraceptives, as well as excessive alcohol consumption. Once the diagnosis of LPL deficiency is confirmed, it is crucial to avoid these secondary causes to effectively manage the condition [19]. In addition to *LPL* and *APOC2*, other less common gene defects can cause monogenic chylomicronemia. *APOC2* ranks as the second most observed cause of monogenic chylomicronemia, while pathogenic variants in other genes like *APOA5*, *LMF1*, and *GPIHBP1* are much rarer [9].

Additionally, it is worth discussing the differential diagnosis of LPL deficiency considering the case of patient 6, who was ultimately diagnosed with GSD1a deficiency. GSD1a deficiency is a rare autosomal recessive disorder due to impaired glucose-6-phosphatase activity, leading to the accumulation of glycogen in various tissues, including the liver and kidneys. Clinical features of GSD1a deficiency include hypoglycemia, hepatomegaly, growth retardation, and lactic acidosis. Some GSD1a patients may also present with hypertriglyceridemia [20], which can obviously mimic LPL deficiency. Pre-prandial lactic acidosis is not part of the clinical spectrum in LPL deficiency, but it is obligatory in GSD1a deficiency. If clinical features do not align with LPL deficiency, the consideration of alternative diagnoses is imperative since therapeutic strategies may differ.

The diagnosis of LPL deficiency in our patients, six children from four families, was confirmed through targeted genetic testing revealing homozygous pathogenic variants in the *LPL* gene in all of them. The high rate of homozygosity highlights the importance of genetic counseling about the increased likelihood conceiving a child with the disorder in consanguineous couples with a family history of LPL deficiency.

The primary treatment goal in LPL deficiency is lowering serum TG levels to prevent pancreatitis. In 2012, gene therapy was developed for a subset of patients with LPL deficiency who were at an increased risk of pancreatitis [21]. The therapy showed promising initial results, including an improved overall quality of life and a decreased risk of pancreatitis [22]. However, despite these positive outcomes, the therapy was discontinued reportedly due to low demand.

Currently, there is no pharmacological treatment available for LPL deficiency, making dietary modification the mainstay of long-term therapy. An LCT-restricted/MCT-supplemented diet has been successfully employed for over a decade at our institution. The success of the MCT diet can be attributed to several factors. MCTs are rapidly absorbed and metabolized, bypassing the impaired lipoprotein lipase function in LPL patients. This allows for the efficient utilization of dietary fats and prevents the accumulation of TGs in the bloodstream. The use of an MCT-enriched formula led to a significant drop in TGs, with values consistently maintained below 20 mmol/L in all cases and some achieving the target range of 10 mmol/L, indicating that MCTs, which are not incorporated into chylomicrons in the small intestine but absorbed directly into the portal vein, can be effectively metabolized as a source of fatty acids without exacerbating hypertriglyceridemia. The LCT restriction component of the diet limits the intake of LCT that would require functional LPL for processing. Technically, this is achieved by replacing LCT with MCT while keeping the total caloric contribution of nutritional fat constant. This treatment approach is used in the management of patients with inherited long-chain fatty acid oxidation defects and has been proven effective and safe [12].

It is important to note that adherence to the prescribed diet plays a crucial role in achieving optimal outcomes. Patients who deviated from the diet or had difficulty maintaining dietary compliance experienced fluctuations in TG levels, emphasizing the need for ongoing education and support for patients and their families. Our patients were all closely monitored and supported by a registered dietitian to help execute the dietary principles in a practical and acceptable manner to the child and family while adjusting the diet through changes in dietary requirements across infancy and childhood. Strategies to support dietary adherence include adjusting the MCT supplement to the taste preferences of the child, education on recipes and low-fat cooking techniques, education on the fat content of foods, and meal planning. In our experience, it is paramount to address dietary compliance with counseling, education, and close monitoring by a multidisciplinary team.

Since the principle of the LCT-restricted/MCT-enriched diet is the same as the ones for the treatment of inherited long-chain fatty acid oxidation defects [12], metabolic dietitians are proficient in calculating and directing the diet for neonates, children, and adults. Special attention is paid to avoiding deficiency in essential fatty acids [12,14]. In our experience, the regular monitoring of fatty acids in the lipid fraction of the erythrocyte membrane helps to identify deficiencies. Dietary adjustments include the supplementation of some plant oils like walnut oil that have a desirable ratio of both essential fatty acids [23].

The patients in this study achieved appropriate weight gain, linear growth, and developmental milestones, and this indicates that the dietary management supports overall health and well-being. The significance of a well-balanced diet is exemplified in the clinical course of patient 5. Diagnosed with elevated TGs in infancy and treated with fat restriction between age 6 and 12, the patient faced challenges in adhering to the low-fat diet. His TGs were persistently elevated, he developed eruptive xanthomata, and he had poor growth and development. After switching to the LCT-restricted/MCT-enriched diet, his metabolic control improved, his eruptive xanthomata resolved, and he had catch-up growth, emphasizing the potential of the LCT-restricted/MCT-enriched diet in managing LPL-related complications, controlling TG levels, and supporting healthy growth.

While the MCT-based diet appears promising, it is important to recognize the challenges associated with long-term adherence. Dietary modifications can significantly impact patients’ lifestyles and may require ongoing support and education to ensure compliance. Moreover, individual variations in response to the diet may exist, as observed in patient 2-I, who experienced fluctuating TG levels despite dietary intervention. This highlights the need for personalized approaches, close monitoring, and regular follow-ups to optimize management outcomes.

Despite the small sample size, our findings contribute to the growing body of evidence supporting the use of dietary interventions, specifically an LCT-restricted/MCT-supplemented diet, as a viable treatment strategy for LPL deficiency. Future studies with a larger sample size, prospective design, and longer follow-up period can provide more robust evidence regarding the efficacy and long-term effects of an LCT-restricted/MCT-enriched diet in the treatment of LPL deficiency.

## 7. Conclusions

The LCT-restricted/MCT-supplemented diet, employed for over a decade at our institution, appears to be amenable for the management of hypertriglyceridemia > 20 mmol/L in LPL patients. Our observations support the feasibility of the diet in the long-term management of children with LPL deficiency and the efficacy of the diet in the prevention of complications associated with hypertriglyceridemia. Prospective cohort studies are necessary to confirm the efficacy of the diet in the long-term management of patients with LPL deficiency.

Notably, all patients in this study presented very early in life and exhibited significant elevations in TGs. This emphasizes the importance of early diagnosis and intervention, which may further improve outcomes in LPL deficiency.

## Figures and Tables

**Figure 1 nutrients-15-03535-f001:**
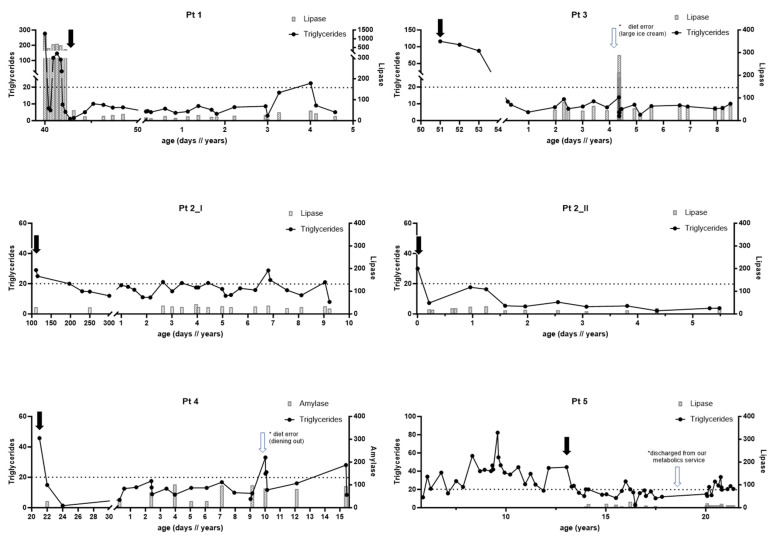
**Triglycerides and pancreatic enzymes in 6 patients with lipoprotein lipase deficiency.** Implementation of LCT-reduced/MCT-enriched diet (black arrows). Please note exceptional events in Pts 3, 4, and 5 (empty arrows).

**Figure 2 nutrients-15-03535-f002:**
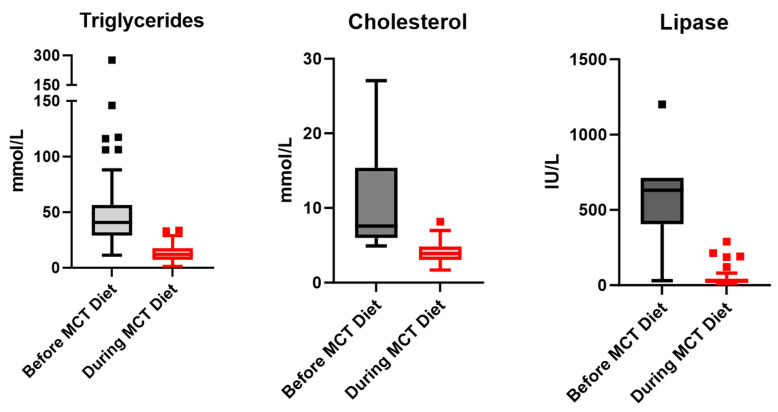
**Triglycerides, cholesterol, and pancreatic lipase before and during treatment with MCT-enriched diet.** Box and Whiskers plot, Tukey method.

**Figure 3 nutrients-15-03535-f003:**
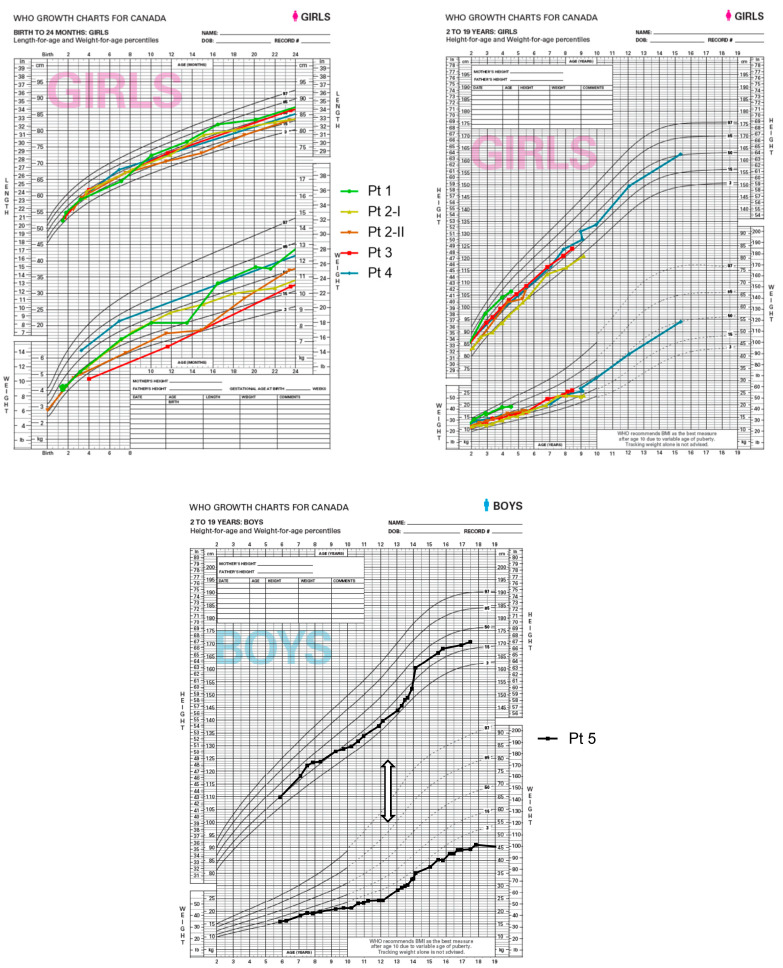
**Growth parameter in 6 patients with lipoprotein lipase deficiency.** Growth recordings began in patients 1–4 at the time of initiation of the LCT-restricted/MCT-enriched diet. Start of diet in patient 5 is marked with the white arrow.

**Table 1 nutrients-15-03535-t001:** Demographics and Clinical Findings.

	Gender	Initial Presentation	Age at Presentation	TG at Presentation	Pancreatitis	Eruptive Xanthoma	Lipemia Retinalis	Genetics	Abdominal Ultrasound	MRI
			[days]	[mmol/L]						
P1	female	Vomiting, diarrhea, irritability, and abdominal distension	39	276	At presentation *	No	No	*LPL*: c.1014 C>A, p.Tyr338* // c.1014 C>A, p.Tyr338*	Demonstrated a mass in the head of the pancreas	Abdominal MRI: Pancreatic inflammation consistent with pancreatitis. Brain MRI: No evidence of infarction or clots
P3	female	Positive family history	51	116	One episode due to diet failure	No	No	*LPL*: c.784 C>T, p.Q262X // c.784 C>T, p.Q262X	Normal	Brain MRI: Normal
P2-I	female	Isolated splenomegaly was noted at a routine well-baby check-up	110	29	No	No	No	*LPL*: c.784 C>T, p.Q262X // c.784 C>T, p.Q262X	Splenomegaly, resolved after initiating diet	Not done
P2-II	female	Positive family history	7	30	No	No	No	*LPL*: c.784 C>T, p.Q262X //c.784 C>T, p.Q262X *ROBO3*: c.1193_1196del, p.T398Sfs*43 // c.1193_1196del, p.T398Sfs*43	Normal	Brain MRI: Brainstem malformation, in keeping with known ROBO3 defect Spinal MRI: Scoliosis
P4	female	Lethargy, poor feeding, irritability, and abdominal distension.	21	46	No	No	No	*LPL*: c.472 G>A, p.A158T // c.472 G>A, p.A158T	Splenomegaly, resolved after initiating diet	Not done
P5	male	Routine workup for circumcision	14	unknown	Chronic pancreatitis*	Yes *	No	*LPL*: c.784 C>T, p.Q262X // c.784 C>T, p.Q262X	Renal ultrasound for bilateral hydronephrosis: normal	Not done
P6	female	Growth failure, hepatomegaly, and respiratory distress	270	113	No	No	No	*LPL*: no variants *G6PC*: c.980_982del, p. F327del // c.563-3 C>G, p.?)	Hepatomegaly	Not done

* No pancreatitis since start of the MCT-enriched diet, TG - triglycerides.

**Table 2 nutrients-15-03535-t002:** LCT-restricted/MCT-enriched diet in LPL patients.

	Diet Start	MCT Supplementation	Vitamin and Mineral Supplements	Essential Fatty Acid Supplements	Average Dietary Intakes						Essential Fatty Acid Levels *		
	Age				MCT	LCT	LA	ALA	EER		LA	ALA	DHA
	[days]				[% of total energy]	[% of total energy]	[% of total energy]	[% of total energy]	[% of EER]		[% of total lipid fatty acid]	[% of total lipid fatty acid]	[% of total lipid fatty acid]
										mean ± SD (range)	9.47 ± 1.04 (7.39–11)	0.097 ± 0.035 (0.027–0.167)	3.35 ± 1.03 (1.29–5.41)
P1	41	<6 months: Lipistart 6 months-2 years: Lipistart, MCT oil >2 years MCT oil, MCT Procal	Vit D	Walnut oil—since age 8 months	19.4	8.4	2.5	0.44	98		8.42	0.17	4.63
P3	51	<1.5 years: Lipistart and Portagen 1.5–5 years: Lipistart and MCT Prolcal >5 years: MCT Procal, MCT oil, Liquigen	<5 years: Phlexy vits >5 years: Vit D, Calcium	Walnut oil—since age 8 months	24.0	8.0	2.7	0.42	106		9.38	0.21	3.28
P2-I	110	<1.5 years: Lipistart >1.5 years: Lipistart, MCT Procal/MCT oil/Liquigen	<2 years: Vit D >2 years: Phlexy vits	Walnut oil—since age 2 years	22.7	5.5	2.5	0.40	104		7.06	0.12	3.75
P2-II	7	<12 months: Lipistart >12 months- currently: Lipistart, MCT oil, MCT Procal	Calcium, Vit D	Walnut oil—since age 8 months	22.3	6.6	2.7	0.41	100		6.77	0.09	4.14
P4	21	<2 years portagen, MCT oil, >8 years stopp portagen switch to MCT Procal	Calcium, Vit D	Walnut oil—since age 2 years	11.4	11.8	2.1	0.33	103		8.34	0.41	3.11
P5	13.4 years	13.5–18 years: MCT Procal, Portagen, Boost Fruit Beverage	Calcium, Vit D, Iron	Walnut oil—since age 13.5 years	17.2	9.4	1.4	0.34	101		9.09	0.36	2.29

* Red Blood Cell Membrane Total Lipid Fatty Acid Profile. Controls: age 8.7 ± 4.6, 0-16 years (mean ± SD, range), n = 44. Abbreviations: TG—triglycerides, MCT—medium chain triglycerides, LCT—long-chain triglycerides, LA—linoleic acid, ALA—alpha-linolenic acid, EER—estimated energy requirement.

## Data Availability

The data presented in this study are available on request from the corresponding author. The data are not publicly available due to privacy restrictions.

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
