# Peer review of "Long-Term Treatment of Lipoprotein Lipase Deficiency with Medium-Chain Triglyceride-Enriched Diet: A Case Series"

_nutrients, 2023, doi:10.3390/nu15163535_

Round 1

Reviewer 1 Report

The manuscript by Aljouda et al. reports a beneficiary effect of medium chain triglyceride (MCT)-enriched diet on managing LPL deficiency, theoretically based on a bypassing of the LPL-mediated lipid absorption. In this retrospective case review study, the authors reported a positive tendency that the MCT diet decreases circulating triglycerides, contains clinical complications, and improves the growth and development of a small cohort of six patients suffering from LPL deficiency. However, the reported results cannot support the author's conclusion due to the fundamental weakness of lacking a control arm and, thus, adequate statistical analysis. The manuscript is not recommended for publication in its current form.

Major concerns:

1. A control group without exposure to the MCT diet must be included to assess the treatment effect scientifically. 

2. Adequate statistical method should be implemented to compare the diet-induced differential patient outcomes.

Author Response

We thank the reviewer for reviewing the manuscript and the critical comments. We hope we could alleviate the reviewers concern with the following adjustments we made.

The manuscript by Aljouda et al. reports a beneficiary effect of medium chain triglyceride (MCT)-enriched diet on managing LPL deficiency, theoretically based on a bypassing of the LPL-mediated lipid absorption. In this retrospective case review study, the authors reported a positive tendency that the MCT diet decreases circulating triglycerides, contains clinical complications, and improves the growth and development of a small cohort of six patients suffering from LPL deficiency. However, the reported results cannot support the author's conclusion due to the fundamental weakness of lacking a control arm and, thus, adequate statistical analysis. The manuscript is not recommended for publication in its current form.

Major concerns:

  1. A control group without exposure to the MCT diet must be included to assess the treatment effect scientifically.
  2. Adequate statistical method should be implemented to compare the diet-induced differential patient outcomes.

Our study is a case series. Performing a controlled study, like a cohort study, is not feasible for the following reason. LPL deficiency is a rare inborn error of metabolism, recruiting enough patients is almost impossible and considering offering no treatment (=control arm) for several years is unethical. We would have liked to include statistical analysis, however the data collected from retrospective chart analysis do not allow meaningful statistical analysis. We calculated median and range of several variables and added them to the text. To avoid any overstatement of our findings and to make sure that our conclusions are supported by the results, we revised our conclusions carefully.

Still from the perspective of a clinician, like me, our observations are convincing and with the help of this publication should encourage other clinicians to treat their LPL patients with the LCT-reduced/MCT-enriched diet.

Reviewer 2 Report

1) title is with acronyms, Please revise. 

2) since this article is a case series, please rewrite the article following the guidelines. 

3) title should contain the statment with case series analysus

4) graphs based on overall analysis should be included without p value

Author Response

We thank the reviewer for reviewing the manuscript and the critical comments. It was especially helpful to be advised on the reporting criteria of case series.

We have

- improved the introduction and revised relevant literature

- revised the description of the methodology

- revised the presentation of the results

1) title is with acronyms, Please revise.

We have revised the title accordingly.

2) since this article is a case series, please rewrite the article following the guidelines.

We thank the reviewer for this helpful criticism. We revised the whole manuscript following the guidance of good practice for case series (Carey, T.S. and S.D. Boden, A critical guide to case series reports. Spine (Phila Pa 1976), 2003. 28(15): p. 1631-4.). Especially we paid attention to the following.

- clearly defined question

- well described study population

- well described intervention

- well described results

- use of validated outcome measures

However, we decided not to use statistical analysis. The data available, especially prior to intervention were scant. Comparing, for example, one pre-treatment value with many treatment values may provide a statistically significant difference, but it is not meaningful. It just overestimates the importance of statistics. Using the clinical judgement instead, is in our opinion more important.

3) title should contain the statment with case series analysus

We have added the case series description to the title

4) graphs based on overall analysis should be included without p value

We have added a new graph (new figure 2) comparing triglycerides, cholesterol, and lipase results pre-treatment and during treatment. We avoided p-values.

Reviewer 3 Report

Summary:  Paper is a case study review of patients with lipoprotein lipase LPL_ deficiency who were treated at the Hospital for Sick Children in Toronto over a 22 year period.  Six patients were identified based on clinical criteria with five ultimately being definitelvy diagnosed with LPL deficiency.  Despite there being no current treatment for this disorder, patients were successfully managed by substituting dietary long chain triglycerides with meals enriched in medium chain triglycerides resulting in good clinical outcomes.

Specific comments:

Major

1.    Page 2, 4th paragraph.  Sentence beginning “However, treatments of LPL patients…”  There are a number of papers that report on treating patients with LPL deficiency including the paper by Williams et al (2018) in the previous sentence.  This sentence should be deleted.

2.    Page 2, First paragraph of Methods.  How many charts were included in the chart review?

3.    Page 3.  First paragraph.  The “lipase” laboratory test mentioned here and elsewhere may confuse some readers who may find it odd that a lipoprotein lipase deficient patient has elevated lipase levels.  It would be better to say pancreatic lipase here and elsewhere when referring the laboratory lipase test.

4.    Page 4.  Second paragraph under Patient 2-I.  Please indicate that the walnut oil is being used as a source of EFA and what the original and increased doses were.  Also, in this paragraph, briefly describe the “unhealthy feeding habits”.

5.    Page 5, top line.  The NAA25 gene.  Please briefly describe the function of this.

6.    It should be mentioned in the discussion that LPL deficiency is one of the major causes of familial chylomicronemia syndrome (FCS).

Minor

1.     Abstract, first sentence.  Please describe the major symptoms that typically develop before age 10 (abdominal pain/pancreatitis).

2.    Glycogen storage disease 1a is abbreviated as GSD, GSD1 and GSD1a in various parts of the manuscript.  Please use a single abbreviation throughout the manuscript.

3.    Page 2, second paragraph.  I think the word in the second sentence of this paragraph is “permanent” rather than “permeant”.

4.    Page 2, third paragraph.  “diacylglycerol acyl transferase 1” should be “diacylglycerol acyltransferase 1 inhibition”.

5.    Page 3.  Last sentence under Red blood cell (RBC) total lipid fatty acid profile. “fatty acid esters” should be “fatty acid methyl esters”.

6.    Page 4, first sentence “LPL diagnosis” should be “LPL deficiency diagnosis”.

7.    Page 4, under patient 2-II.  Sentence “TGs were 30 mmol/L confirming her diagnosis of LPL deficiency”.  I think it would be more accurate to say “consistent with” rather than “confirming” since it’s possible that there are other reasons for the hypertriglyceridemia.  The genetic workup confirmed the diagnosis.

8.    Page 5.  I don’t really see the need to list study limitations since this was simply a chart review and not a clinical trial.  If the paper needs to be shortened this paragraph can be deleted.

9.    Page 10 (Table 2).  The footnote to the table lists Red Blood Cell Membrane Total Fatty Acid Profile but I don’t see that in the table.  There is also a footnote about Controls but I don’t see any controls in the table either.  These should be deleted if they don’t correspond to anything in the table.  Also, include DHA in the list of abbreviations for this table.

10. Page 12.  The heading “Growth and EFF” should be “Growth and EFA”.

11. Page 13.  Bottom paragraph. Sentence beginning “In addition to LPL and apoC-II…”.  Did you mean apoC-III instead of apoC-II?  The gene name for apoC-III is APOC3.

12. Page 13, last sentence.  I don’t understand what “LPL/LMF1” or “LPL/ApoA5” are.  I don’t know why these are listed since LMF1 and APOA5 are already shown here.

13. Page 14. First sentence.  “GSD1a” should be “GSD1a deficiency”.

14. Page 14, second paragraph. Sentence beginning “The diagnosis of LPL deficiency…”  The paper reports on six children with LPL deficiency from five families rather than five children from four families.

There are some minor corrections that I have pointed out in the comments to authors.

Author Response

Summary:  Paper is a case study review of patients with lipoprotein lipase LPL_ deficiency who were treated at the Hospital for Sick Children in Toronto over a 22 year period.  Six patients were identified based on clinical criteria with five ultimately being definitelvy diagnosed with LPL deficiency.  Despite there being no current treatment for this disorder, patients were successfully managed by substituting dietary long chain triglycerides with meals enriched in medium chain triglycerides resulting in good clinical outcomes.

We thank the reviewer and are grateful for the thorough review and helpful comments.

Specific comments:

Major

  1. Page 2, 4th paragraph. Sentence beginning “However, treatments of LPL patients…”  There are a number of papers that report on treating patients with LPL deficiency including the paper by Williams et al (2018) in the previous sentence.  This sentence should be deleted.

Thank you. We rewrote this sentence referring to the one study using MCT.

  1. Page 2, First paragraph of Methods. How many charts were included in the chart review?

Seven charts were included in the chart review.

  1. Page 3. First paragraph.  The “lipase” laboratory test mentioned here and elsewhere may confuse some readers who may find it odd that a lipoprotein lipase deficient patient has elevated lipase levels.  It would be better to say pancreatic lipase here and elsewhere when referring the laboratory lipase test.

Thank you, that’s an excellent point. We replaced lipase with pancreatic lipase throughout the text.

  1. Page 4. Second paragraph under Patient 2-I.  Please indicate that the walnut oil is being used as a source of EFA and what the original and increased doses were.  Also, in this paragraph, briefly describe the “unhealthy feeding habits”.

We added the information.

  1. Page 5, top line. The NAA25 gene.  Please briefly describe the function of this.

Done.

  1. It should be mentioned in the discussion that LPL deficiency is one of the major causes of familial chylomicronemia syndrome (FCS).

Done. Thank you.

Minor

  1. Abstract, first sentence. Please describe the major symptoms that typically develop before age 10 (abdominal pain/pancreatitis).

Done. Thank you.

  1. Glycogen storage disease 1a is abbreviated as GSD, GSD1 and GSD1a in various parts of the manuscript. Please use a single abbreviation throughout the manuscript.

Done.

  1. Page 2, second paragraph. I think the word in the second sentence of this paragraph is “permanent” rather than “permeant”.

Thanks for spotting the typo. Corrected.

  1. Page 2, third paragraph. “diacylglycerol acyl transferase 1” should be “diacylglycerol acyltransferase 1 inhibition”.

Corrected. Thank you.

  1. Page 3. Last sentence under Red blood cell (RBC) total lipid fatty acid profile. “fatty acid esters” should be “fatty acid methyl esters”.

Corrected, thanks.

  1. Page 4, first sentence “LPL diagnosis” should be “LPL deficiency diagnosis”.

Done.

  1. Page 4, under patient 2-II. Sentence “TGs were 30 mmol/L confirming her diagnosis of LPL deficiency”.  I think it would be more accurate to say “consistent with” rather than “confirming” since it’s possible that there are other reasons for the hypertriglyceridemia.  The genetic workup confirmed the diagnosis.

Agree. Changed, thanks.

  1. Page 5. I don’t really see the need to list study limitations since this was simply a chart review and not a clinical trial.  If the paper needs to be shortened this paragraph can be deleted.

I agree, that’s a good point. I removed one sentence and re-arrange the last paragraph of the discussion.

  1. Page 10 (Table 2). The footnote to the table lists Red Blood Cell Membrane Total Fatty Acid Profile but I don’t see that in the table.  There is also a footnote about Controls but I don’t see any controls in the table either.  These should be deleted if they don’t correspond to anything in the table.  Also, include DHA in the list of abbreviations for this table.

Thank you again for spotting that we the first row of the table was missing. It is now corrected.

  1. Page 12. The heading “Growth and EFF” should be “Growth and EFA”.

I spelled it out as Essential Fatty acids.

  1. Page 13. Bottom paragraph. Sentence beginning “In addition to LPL and apoC-II…”. Did you mean apoC-III instead of apoC-II?  The gene name for apoC-III is APOC3.

apoC-II is correct. Following LPL deficiency, APOC2 pathogenic variants represent the second most reason for monogenic chylomicronemia. apoC-III is a therapeutic target for chylomicronemia.

  1. Page 13, last sentence. I don’t understand what “LPL/LMF1” or “LPL/ApoA5” are. I don’t know why these are listed since LMF1 and APOA5 are already shown here.

That was put in in error and has been corrected. Thank you.

  1. Page 14. First sentence. “GSD1a” should be “GSD1a deficiency”.

Done.

  1. Page 14, second paragraph. Sentence beginning “The diagnosis of LPL deficiency…” The paper reports on six children with LPL deficiency from five families rather than five children from four families.

Thanks for spotting the typo. It has been changed to six patients from four kindreds. Patient 3 is cousin of siblings P2-I and -II.

Thank you again for the thorough review and valuable comments.

Round 2

Reviewer 2 Report

no additional comments

the paper fits the requirements for the publication in this current format.

all authors addressed my comments.